# Citizens' perspectives on relocating healthcare

**L. J. Damen**[1]*, **J. D. De Jong**[1,2], **L. H. D. Van Tuyl**[1], **J. C. Korevaar**[1]

**1** Nivel, the Netherlands Institute for Health Services Research, Utrecht, the Netherlands, **2** CAPHRI, Maastricht University, Maastricht, the Netherlands

\* l.damen@nivel.nl

## Abstract

### Background

Healthcare systems around the globe are facing challenges. There are increasing demands and costs at the same time as a diminishing health workforce. Without change, healthcare will become unsustainable. The Dutch government is searching for solutions, one of which is relocating healthcare. Relocating healthcare from expensive institutions to sites closer to patients' homes is an important part of this. This relocation is expected to reduce costs and lessen shortages of personnel. However, although citizens have an important stake in this, little is known about how they think about this topic. This research aims at investigating citizens' perspectives on relocating care.

### Methods

In December 2021, three open-ended questions were sent to 1,500 members of Nivel's Dutch Healthcare Consumer Panel, 796 respondents responded. In addition, two citizen platforms were organised in March and April 2022. A total of 23 citizens participated.

### Results

Our results indicated that the following aspects are important for citizens in healthcare delivery: being treated by someone with expertise in the area of their need, familiarity with the healthcare provider and the treatment of less complex care close to home. When certain conditions are met, citizens prefer treatment for less complex care from their general practitioner rather than in a hospital. The most important condition is that the general practitioner has the right expertise regarding their health question. The willingness to relocate care from the general practitioner to other healthcare providers or to self-care is also present. One of the problems, however, is that citizens often do not know to which healthcare provider they should go or what they should do to increase self-care.

### Conclusion

From a citizens' perspective, relocating care is an acceptable solution for keeping healthcare sustainable in the future, provided that certain conditions are met.

**Data Availability Statement:** The qualitative data collected and analysed during the current study is not publicly available, but is available on reasonable request from the authors and is subject to approval by the program committee of the Dutch Healthcare Consumer Panel. This program committee

supervises processing the data of the Dutch Healthcare Consumer Panel and decides about the use of the data. The program committee consists of representatives of the Dutch Ministry of Health, Welfare and Sport, Health and Youth Care Inspectorate, Zorgverzekeraars Nederland (Association of Healthcare Insurers in the Netherlands), the National Healthcare Institute, the Federation of Patients and Consumer Organisations in the Netherlands, the Dutch Healthcare Authority and the Dutch Consumers Association. All research conducted within the Consumer Panel has to be approved by this program committee. The committee assesses whether a specific research fits within the aim of the Consumer Panel, that is strengthen the position of the healthcare user. Data are available upon request from rvb@nivel.nl, the executive board of Nivel, under the name "ALG-017 Right Care in the Right Place".

**Funding:** The author(s) received no specific funding for this work.

**Competing interests:** The authors have declared that no competing interests exist.

## Introduction

There is an increasing realisation at a global level that without reorganisation healthcare systems are unsustainable [1–3]. Simultaneous developments such as population ageing, advances in medicine, and rising incomes all lead to an increase in demand for healthcare. With a rise in demand, costs will also increase. Even in a scenario where governments are able to contain costs, it is expected that spending on healthcare will increase further from 6% of the gross domestic product in 2015 to 8,2% in 2030 and to 9,5% in 2060 across the Organization for Economic Co-operation and Development (OECD) countries [4]. If governments are unsuccessful in containing costs, public spending is projected to be even higher, namely 14% of the gross domestic product by 2060. Another problem in the organisation of care is the shortage of healthcare workers. By 2030, there is expected to be a net shortage of 15 million health workers worldwide; in 2013 this shortage was seven million [5]. Therefore, sustainable solutions are being sought in various countries. Governments are searching for ways to organise healthcare in such a way that it is affordable, accessible, and still delivers care of high quality [2, 3, 6]. When in the process of organising care differently, governments could stimulate the improvement of the efficiency of healthcare. This could be achieved, for example, by improving the coordination of care, stimulating the transition to more digital healthcare, relocating healthcare, for example, from hospital care to primary care, or by investing in health promotion and disease prevention [7].

In some countries, such as the United Kingdom, Austria, and China, governments are focussing on integrated care, involving multidisciplinary teams, integration across sectors, and rehabilitating patients at home. This has resulted in an improved coordination of care and a decrease in the burden on the system [8]. In the Netherlands, one of the programmes to enhance the sustainability of the healthcare system is called: '*the Right Care in the Right Place*'. Several parties in the healthcare field collaborate within this initiative to firstly prevent healthcare or more expensive healthcare, for instance by prevention, secondly to relocate healthcare, for example, closer to people's homes, and thirdly to replace healthcare, for instance, with a system of eHealth [9]. This study focused on relocating healthcare.

Research has shown that in the Netherlands 46% of all care activities provided in a hospital could be relocated and provided at home within ten years. This figure was achieved by analysing which type of activity could be safely performed at home with currently existing technology [10]. Moreover, based on pilot studies, estimates, have been made that the relocation of hospital care could reduce costs by €1,5 billion [11]. This would be achieved by relocating care to the general practitioner (GP), to one-and-a-half line clinics (in which a specialist performs treatments in the general practice), to primary care centres, or to self-care.

It is important to gain insight into the perspectives and needs of citizens in order to relocate care. Several studies have looked into the subject of patients' preferences for relocating care [12–20]. However, most are quantitative studies [12–17] and/or relate to a very specific study population [17, 19, 20], or to a very specific form of relocating such as to one-and-a-half line care [14–16]. What is missing is a qualitative insight into citizens' preferences and needs relating to relocating care. These insights will increase our understanding of the acceptance and needs of citizens. Such understanding can help to improve the organising of this relocation of care and raise the quality of the care. These qualitative insights are the focus of this study.

This study aimed to explore the perspectives of citizens on relocating care. We looked, in particular, at aspects that are important in health provision, and also at the conditions required to relocate care either from the hospital to the general practice or from the general practice to other healthcare providers (HCPs) or self-care. We first examined what citizens consider important in healthcare provision as this is highly relevant to the way in which the relocation

of healthcare is organised. Secondly, we examined the conditions that are important to citizens for relocating care. The following research question was formulated: what are the perspectives of citizens on relocating healthcare? To answer this question two subsequent questions were drawn up which were:

- What are the perspectives of citizens on *the Right Care in the Right Place*, when it comes to what is important in healthcare provision?

- What conditions are important to citizens when relocating care?

## Method

### Design

This study employed two distinct methods to address the research questions. Firstly, a questionnaire with three open-ended questions was distributed to citizens to achieve broad participation. Secondly, two citizen platforms were organized to facilitate in-depth discussions.

The questionnaire was chosen for its ability to reach a large audience, while the citizen platforms allowed for more detailed engagement with a smaller group. The citizen platforms are designed to involve participants in discussions on complex health issues, which they might otherwise find difficult to engage in. In the Netherlands, this method was developed by Nivel, the Netherlands Institute for Health Services Research [21], and was inspired by the UK National Institute for Health and Care Excellence Citizens Council [22]. The method features a one-day program where citizens participate in interactive assignments and attend expert presentations to develop informed opinions. The goal is not to reach a consensus but to capture a range of perspectives.

### Sample and recruitment

The questionnaire was sent to 1,500 panel members of the Dutch Healthcare Consumer Panel in December 2021. This Panel, managed by Nivel, collects information on a national level on the opinions, knowledge, expectations, and experiences of healthcare users [23]. In 2021 the panel consisted of more than 11,000 people aged 18 years and older. Inclusion criteria are that people must be aged 18 years and older, live in the Netherlands, and are able to understand Dutch. There are no exclusion criteria. Background characteristics from all panel members, such as age, gender, ethnicity, and highest level of education completed, are assessed at the start of the panel membership. The sample of the survey was representative of the general population by age and gender. Panel members who did not respond were sent two reminders. 796 (53%) panel members responded to the questionnaire.

The participants for the citizen platforms were also recruited through the Dutch Healthcare Consumer Panel. Based on previous experiences with citizen platforms, 20–30 participants are deemed optimal for a successful citizen platform. An invitation was sent on 23 February 2022 to a sample of 2000 panel members who were representative of the general population by age and gender. Of these, 48 people expressed interest in participating. Subsequently, 20 citizens were selected based on the following characteristics: gender, age, education, whether or not they had a chronic condition, and the number of different healthcare providers the person had contact with in the past six months. This selection aimed to create a diverse group. After confirming participation, some cancellations occurred, and replacements were sought from other interested panel members with similar background characteristics. Eventually, sixteen people attended the citizen platform on 25 March. Two of the 20 people invited did not attend, for

unknown reasons, one participant could not make it due to transport problems, and one tested positive for Covid-19.

Recruitment for the second citizen platform followed a similar approach. However, this time participants were chosen based on their interest of being involved with *the Right Care in the Right Place*. In an earlier survey panel members were asked to what extent panel members wanted to be involved in *the Right Care in the Right Place* [24]. We expected that those wanted to be involved with *The Right Care In The Right Place* might have different perspectives on relocating care than those less interested to be involved. All panel members who did not wanted to be involved (n = 302) were invited. The sample was supplemented with young people (aged 18–39) and low-educated people, those characteristics were associated with not wanting to be involved [24]. On 28 February an invitation was sent to 1,022 panel members, of whom 17 wanted to participate and ten were invited. Seven participants attended on 1 April. Three invitees did not attend. One of them did not have internet at home, one only wanted to participate in a physical meeting, and one dropped out for an unknown reason. Four people from the second citizen platform had also completed the questionnaire.

## Data collection

**Questionnaire.** Three open-ended questions about *the Right Care in the Right Place* were included in the December 2021 survey (Appendix A in S1 File). The purpose of the questions was to examine what is important within healthcare provision/*the Right Care in the Right Place* according to citizens. In addition, the answers provided input for the citizen platforms scheduled for spring 2022.

**Citizen platforms.** Citizen platforms were organised on 25 March, located at Nivel, and online at 1 April, 2022. The objective of these platforms was to examine citizens perspectives on relocating care. The programme was compiled and performed by researchers from Nivel. To compile the program, we utilized information gathered during an exploratory focus group with GPs [25], which helped us decide which forms of relocation we wanted to discuss.

The researchers involved had diverse positions (junior researchers, senior researchers, and executives), varying years of experience (ranging from 1 to 15 years), and different backgrounds (including: sociology, economics, science and innovation management, epidemiology, and occupational therapy). During the citizen platforms, the researchers' role was to moderate and observe.

The days consisted of different activities with breaks in between, in accordance with the Nivel Platform guidelines [26]. Appendix B in S1 File shows the time schedules of the citizen platforms. Presentations were held by Nivel and the Ministry of Health, Welfare and Sport, to explain more about the challenges of the healthcare system and about *The Right Care In The Right Place*. Moreover, several working methods were used (Table 1). During the working sessions, participants were presented with cases of care that could be relocated (from the hospital to the GP or from the GP to other HCPs/self-care), helping them visualize the situation. These cases were compiled and drafted in collaboration with a GP to ensure realism. Participants brainstormed their thoughts on these cases, considered other forms of care that could be relocated, and discussed the conditions under which such relocations could occur. For more detailed information on the programme, see Appendix C in S1 File.

The citizen platform on 1 April was shorter than on 25 March because only the questions that had not been sufficiently answered on 25 March, as discussed by the research team, were asked again. The citizen platform on 25 March took place at Nivel. Discussions in smaller groups (component C, Table 1) were audio-recorded. Additionally, participants documented the findings of each group or individual on paper or flip charts, while Nivel researchers also

**Table 1. Working methods citizen platform 1–25-03-2022.**

| Component—Topic | Working methods | |
|---|---|---|
| A | Important aspects in care delivery. | To examine what citizens consider important in care delivery, participants received five cards with different aspects of healthcare. They were asked to rank these aspects in order of importance. A sixth card was provided, which they could use to write their own aspect. The results were then discussed during a plenary session. |
| B | Associations with the *Right Care in the Right Place* | During a plenary session a word cloud was made with aspects that came to mind when the participants were thing of *The Right care in The Right Place*. |
| C | The relocation of secondary care to primary care | Participants were provided with cases (Appendix C in S1 File) that included scenarios where care could be relocated from the hospital to the GP. There were three groups, each assigned a different case. In these smaller groups participants discussed their preferences for receiving care at the hospital or the GP, along with their reasons. They were also asked to provide examples of care that could be relocated from the hospital to the GP. |
| D | Relocating care from the GP to other Healthcare Providers (HCPs) | Participants were presented with two cases (Appendix C in S1 File) in which sought help from a HCP. The cases described the complaints but did not specify which HCP to consult. In the Netherlands, HCPs in primary care are directly accessible without a referral, allowing citizens to choose to which HCP to visit. Participants were asked what they would do in each situation and to which HCP they would go. They wrote their answers on post-it notes and placed them on a flip chart. The responses were then organized and discussed collectively, with participants explaining the reasons behind their choices. |
| E | Relocating care from the GP to self-care | To examine citizens' thoughts about relocating care from the GP to self-care four cases were presented (Appendix C in S1 File). Participants were asked whether they would go to the GP or not, using a green card to indicate they would go to the GP and a red card to indicate they would not. After this, they were asked to explain why they chose that option and, if they chose not to go to the GP, whether they would take any alternative actions. This was done plenary. Citizens were also asked for conditions to relocate care to self-care. |

*In the left column, each working form is indicated by a letter for easy reference throughout the rest of the article. This can be seen in the column labeled "Component."

*A citizen platform features a one-day program where citizens participate in interactive

took notes. The subsequent citizen platform on April 1 was conducted online, allowing for the audio recording of all the discussions, in addition to the documentation made by participants and notes taken by the researchers.

## Data analysis

**Questionnaire.** First of all, fully missing cases were removed from the database,792 cases remained. The results of the open-ended questions were analysed using thematic analysis [27]. Initially, the results were merged into one file. As an initial step to aid familiarisation, this merged file was read and re-read, before coding the document by highlighter. In addition, the

results were themed, see Appendix D in S1 File for an overview of all the themes. After defining and naming the themes they were interpreted, discussed with the team, and reported.

**Citizen platforms.** The audio records that were made during the citizen platforms were transcribed by L.D.. In addition, the notes that were made by the Nivel researchers were compared and checked for contrary interpretations, parts that were unclear or missing. In such cases, the section was discussed or listened to further until the researchers came to an agreement. Furthermore, the written assignments and flip charts created by participants were cross-referenced with the audio recordings and researchers' notes to verify the results and add any remarks about information not captured in the audio or notes. Finally, the notes, written assignments and flip charts, and transcriptions were combined, merged into one file and summarised. The results were again discussed within the research team. This process ensured that data analysis was triangulated from different perspectives to minimise bias. Moreover, to enhance the credibility of our research, we carried out 'peer debriefing' with a group of peer researchers who were not involved in the study. Furthermore, a member check was conducted by sending a summary of the results to the participants. Participants were invited to respond if they believed something was incorrect or missing from the summary. In this way, the findings and interpretations were tested with the participants.

## Ethical procedures

Data of the questionnaire were analysed anonymously and the privacy of the panel members was guaranteed. This is described in the privacy policy of the Dutch Healthcare Consumer Panel. This complies with the General Data Protection Regulation (GDPR) [28]. According to Dutch legislation, neither obtaining informed consent, nor approval by a medical ethics committee, is obligatory for conducting research through the panel [29].

All participants in the citizen platforms signed a consent form or confirmed their consent verbally on recording at the start of the citizen platform giving Nivel permission to use the data from the citizen platform.

## Results

### The characteristics of the participants

Most respondents to the questionnaire were aged 40 to 64 (50%), 48% were highly educated, most of them used little care (38%), and around 57% of the respondents had no diseases or disorders (See Table 2 for further characteristics of the respondents of the questionnaire).

Most people participating in the citizen platforms were female (13 out of 23 participants), 15 out of 23 participants were highly educated, most participants had contacted a HCP one or two times during the last six months and 14 participants had a chronic disease (Table 3).

### What are the perspectives of citizens on *the Right Care in the Right Place,* with regard to what is important in healthcare provision?

**The expertise of the healthcare provider is seen as most important.** Participants of the citizen platform identified expertise as the most crucial aspect of healthcare delivery (Table 4). They, along with respondents from the questionnaire, frequently associated this expertise with the concept of *the Right Care in the Right Place.* When healthcare professionals possess strong expertise, citizens are more willing to travel further for care.

A key topic of discussion was the specialization of hospitals in particular aspects of care. The majority of questionnaire respondents and citizen platform participants agreed that hospitals should not have too many specializations under one roof, as this can compromise quality.

**Table 2. Background characteristics questionnaire among members of the Dutch Healthcare Consumer Panel (N = 792).**

|  | Proportion |
|---|---|
| **Gender** |  |
| Male | 50,9% |
| Female | 49,1% |
| **Age** |  |
| 18–39 | 20,5% |
| 40–64 | 49,8% |
| 65 and older | 29,8% |
| **Education level** |  |
| Low | 8,1% |
| Middle | 44,0% |
| High | 47,8% |
| **Use of care** |  |
| None | 9,0% |
| Very little | 32,4% |
| Little | 37,6% |
| Much | 18,5% |
| Very much | 2,5% |
| **Chronic disease** |  |
| No | 57,4% |
| Yes | 42,6% |

**Table 3. Background characteristics of citizen platforms N = 23.**

|  | Citizen platform 1 (n = 16) | Citizen platform 2 (n = 7) | Total (N = 23) |
|---|---|---|---|
|  | **Number** | **Number** | **Number** |
| **Gender** |  |  |  |
| Male | 6 | 4 | 10 |
| Female | 10 | 3 | 13 |
| **Age** |  |  |  |
| 18–39 | 3 | 5 | 8 |
| 40–64 | 8 | 0 | 8 |
| 65 and older | 5 | 2 | 7 |
| **Education level** |  |  |  |
| Low | 1 | 1 | 2 |
| Middle | 4 | 2 | 6 |
| High | 11 | 4 | 15 |
| **Number of healthcare providers contacted in the last 6 months** |  |  |  |
| None | 4 | 1 | 5 |
| 1–2 | 4 | 5 | 9 |
| 3–5 | 6 | 1 | 7 |
| 6–10 | 1 | 0 | 1 |
| **Chronic disease** |  |  |  |
| No | 5 | 4 | 9 |
| Yes | 11 | 3 | 14 |

**Table 4. Participants' ranking of aspects important in healthcare provision (N = 23) (citizen platforms–component A, (Table 1)).**

| Aspect | Number of times 1st place | Number of times last place | Average position (range 1–6) |
|---|---|---|---|
| *Expertise* | 18 | 0 | 1,40 |
| *Familiarity with the caregiver* | 3 | 8 | 3,25 |
| *Waiting time* | 1 | 0 | 3,45 |
| *Individual choice** | 1 | 3 | 3,70 |
| *Costs for the patient* | 2 | 4 | 4,30 |
| *Distance* | 0 | 8 | 4,60 |

*Each participant could add one aspect of their choice at a blank card.

*One of the assignments during the citizen platform involved ordering cards with various aspects of healthcare delivery by importance. Each participant received five cards with predefined aspects and one blank card to add their own aspect. The average ranking position for each aspect was calculated from the 23 individual rankings, and the results are shown in this table.

However, a minority believed that every hospital should offer most types of specialty care. Participants with chronic illnesses supported this view, explaining that distance and travel time are significant concerns for them due to their frequent need for specialized care. They preferred having these specializations available close to home or in a single location.

> *"I had expertise at number one. I do not use care much. If I use care, I want to be helped well. I prefer to be helped well, in one go."* [Citizen platform 2 –participant 18]

After expertise, participants of the citizen platform identified familiarity with their healthcare providers (HCPs) as the most important aspect of care delivery. Some participants emphasized that face-to-face contact is crucial for establishing familiarity, while others preferred having low-threshold contact options, such as eHealth.

However, despite the overall importance placed on familiarity, eight participants ranked it as the least important aspect. Some respondents who rated familiarity low still valued the relationship between the patient and HCP, prioritizing aspects such as equality between the patient and HCP, effective communication, patient-centered care, and shared decision-making. Patient-centered care, was also commonly cited by questionnaire respondents as integral to the concept of the *Right Care in the Right Place*.

> 'I think familiarity with the HCP is the most important thing. If the relationship is bad, then neither do you trust what they prescribe' [Citizen platform 2, participant 20]

> 'I have face-to-face contact as an extra aspect. I realised I missed this during the Covid-19 pandemic. I do not like it when the assistant asks me on the phone what I want to make an appointment for. The assistant has nothing to do with that [Participant 9]. I do miss the low threshold contact. It takes a long time before I go to the GP. I would like to have low-threshold contact. I would like to ask: this is bothering me, should I be worried? I do not want to see a GP right away, but I do want to get some guidance to my question [Participant 1].' [Citizen platform 1]

**Distance and costs are seen as least important.** Participants of the citizen platforms considered distance and costs to be the least important aspects of care delivery. They argued that

distance is especially insignificant when there is no urgency or when seeking more complex care. Participants expressed a willingness to pay more or travel further for high-quality or complex care. Additionally, some participants with chronic illnesses noted that they always meet their deductible, the amount paid out-of-pocket before health insurance coverage begins (excluding GP care). Therefore, the cost of care is not a significant concern for them.

*"No, I do not use care that much. Maybe if you used it more you would consider distance more important. But if you ask me now, I'd say rather further away than less good care."* [Citizen platform 2 –participant 17]

Regarding *the Right Care in the Right Place*, respondents to the questionnaire often mentioned accessibility, by which they mostly meant care close to home or provided at home. Participants from the citizen platform focused mainly on the need for non-complex care to be available close to home.

### What are the conditions for relocating healthcare according to citizens?

Relocating care was thought to be a good idea according to most of the participants of the citizen platforms and among the respondents to the questionnaire who mentioned this. However, a smaller group disagreed. Respondents from the questionnaire argued for example that GPs lack the expertise to take over hospital tasks and that everyone should stick to their specific roles. A participant from the citizen platform expressed skepticism about cost savings from care relocation and suggested that higher-income citizens should pay more under a reformed funding system. Among those who supported relocating care, there were conditions attached to their approval (Fig 1).

**Conditions for relocating care from the hospital to the GP.** *The right expertise.* The most important condition for relocating care from the hospital to the GP, according to participants of the citizen platforms, was that the GP possesses the right expertise and experience. Other conditions mentioned include good communication between the GP and the specialist, sufficient resources, the financial means, and the availability of personnel. Furthermore, the GPs should not be overburdened by relocating care. As an alternative, it was suggested that the medical specialist could carry out treatments within the general practice. Respondents from the questionnaire mentioned relocating care from the GP to the doctor's assistants to reduce the workload.

| **Conditions for relocating care from the hospital to the GP** | **Conditions for relocating care from the GP to other HCPs** | **Conditions for relocating care from the GP to self-care or social services** |
|---|---|---|
| • The GP has the right expertise<br>• There is good communication between GP and specialist<br>• There are sufficient recources<br>• The GP should not be overburdened | • Knowing which HCP to contact | • Citizens must have the right knowledge<br>• Attention for prevention<br>• Attention for strengthening networks |

**Fig 1. Overview of conditions named for relocating care during the citizen platforms.**

Participants named the following care as care that could be relocated: less complex care; infusions; injections; dialysis; the treatment of chronic patients by a practice nurse such as those who suffer from chronic obstructive pulmonary disease (COPD), diabetes, or mental health problems; minor surgery, such as removing warts; testing blood; removing stitches; wound care; and blood pressure tests.

'*Some things are done very rarely by GPs. Or at least that is my assessment. Some things are done very often by a hospital. When it comes to the human body, if they do not have the skill, then the risk is just too high. It depends on the routine. Experience is important, as well as knowing how to do it.*' [Citizen platform 1 –participant 13]

'*Hospital in miniature*', *I wrote this down to provoke thought. If we are going to adapt a general practice to a hospital in terms of treatments, are you not going to shift the costs to the general practice*?' [Citizen platform 2 –participant 17]

**A preference for going to the GP.** When the conditions are met, most of the participants of the citizen platforms, except for two, preferred going to the GP with less complex care instead of going to the hospital. The reasons mentioned were: familiarity with the GP; lower costs for the patient and society; a shorter distance to the GP than to the hospital; and going to the GP takes less time. Some participants also conducted a risk assessment, preferring to go to the GP if the likelihood of complications was low. The two participants that preferred to go to the hospital had negative experiences with their GP in the past, and, therefore, did not trust the GP's expertise.

'*There is no risk, except that it may have to be done again. You run no risk, nothing could happen. At the most, you get no result. So this is the type of examination that I would like to see my GP carry out.*' [Citizen platform 1 –participant 3]

'*I would not know why I would go to the hospital for that. If I go to the GP with this (then) they can do bigger and more complex care in the hospital.*' [Citizen platform 1 –participant 15]

'*I also think all GPs have been trained, so there is no reason why the expertise would not be there and so there is no reason why I should go to the hospital.*' [Citizen platform 1 –participant 9]

**Conditions for relocating care from the GP to other HCPs.** *Knowing which HCP to contact*. According to participants of the citizen platforms, the most important condition for relocating care from the GP to other HCPs, such as a physiotherapist or a dietician, is ensuring that citizens know which professional to contact for their specific needs. This sentiment was echoed by respondents to the questionnaire. Many participants of the citizen platforms mentioned they were unaware of having direct access to a physiotherapist or dietician. Also, some participants did not know which HCP to contact, and so they would first get a referral from their GP. To avoid unnecessary GP visits, some participants suggested contacting their health insurer or the doctor's assistant to determine which HCP would be best to consult for their specific needs.

The most commonly mentioned reasons for going directly to another HCP instead of the GP were time savings and low risk. However, some participants did see risks, arguing that GPs

have a holistic view of their health, which other HCPs might lack, potentially leading to something being overlooked. These participants preferred to consult their GP as a first step.

*'You can go straight to the physio and he will tell you what to do. And then you have taken burden off the GP.'* [Citizen platform 1 –participant 1]

**Conditions for relocating care from the GP to self-care or social services.** *Citizens having the right knowledge.* An important condition required to increase self-care among citizens, is, according to participants of the citizen platforms, that citizens have the necessary knowledge. Participants mentioned that people need to learn more about self-care, suggesting that education on this topic should begin in schools. Furthermore, participants argued that reliable information on self-care should be available, presented on a website, portal, or app. Participants also stated that citizens should have easy access to professionals. For example, a website could allow professionals to advise whether, and which, healthcare provider or social service citizens should visit.

Another condition mentioned is that there should be more attention to prevention and to strengthening networks in society so that people can seek help from their network rather than going straight to a GP. Finally, participants indicated the GP should always remain available, self-care should not replace the GP.

*'You have to be able to ask your questions somewhere. Then you know whether you need to go to the GP or to the physio, for example. I don't think everyone knows where you can go.'* [Citizen platform 1 –participant 1]

*'Nowadays, we run to the GP. I have had a headache for three days. How come*? *We used to solve it a lot more with grandma.'* [Citizen platform 1 –participant 12]

Most participants agreed on the fact that self-care could be applied with minor ailments. The ailments most referred to were abdominal pain, colds, fever, headache, and diarrhoea.

## Discussion

This study aimed at describing the perspectives of citizens on relocating care. This was achieved by means of three open-ended questions in a questionnaire and through two citizen platforms.

Our results indicate that several aspects are important for citizens in healthcare delivery. Firstly, they stressed being treated by someone with expertise in the area of their need. Secondly, familiarity with the HCP was seen as important, as was treatment of less complex care close to home. There is a high willingness to travel further or pay extra if it contributes to the quality or expertise of care. For less complex care to be relocated from the hospital to the GP, citizens named several conditions, including that the GP should be experienced, there must be good communication between the GP and specialists, and sufficient personnel and financial resources should be available at the general practice. Citizens are also willing to relocate care from the GP to other HCPs or self-care. However, there is a lack of knowledge among citizens about which HCP to visit and what they should do to increase self-care. This must be clear when care is being relocated. Other conditions mentioned for relocating care from the GP to self-care were that there should be more attention to prevention and strengthening networks in society. Participants mentioned that this would enable people to ask each other for healthcare advice more often. Citizens prefer self-care in the treatment of minor ailments over going to their GP.

The finding that familiarity with the caregiver and the requirement of a level of expertise are important in care delivery, and have a major influence on the attitudes towards relocating care, is consistent with the literature [13, 18, 30, 31]. The literature indicates that expertise and familiarity with the caregiver are important factors for patients assessing the quality and trustworthiness of care [30, 31]. However, there were also eight participants in this study who thought that familiarity was the least important aspect of healthcare provision, indicating that it may not be significant for everyone. There might be a difference between people who use care frequently and those who use it less often.

In other studies on relocating care from the hospital to the GP, the main reason patients do not want to relocate care is the fear that GPs have insufficient knowledge [13, 18]. In addition, Wildeboer et al. (2018) reported that patients in their study had a better relationship with their specialist than with their GP, and were, therefore, not willing to relocate care [18]. The reason for this attitude towards relocating is the same as found in this research. However, citizens in this study were more familiar with the GP instead of with the specialist. This may indicate that it is difficult to relocate patients to their GP who have already been receiving hospital treatment for some time, but it remains a viable option for new patients.

Consistent with literature is that citizens prefer to be treated by a GP for less complex care [12, 13]. And that they want to treat minor ailments themselves [32, 33].

It is assumed that reforming healthcare systems with the aim to make better use of resources will make a key contribution to keeping healthcare sustainable. One way to do this is by relocating care to more cost-effective settings [7]. To organise the relocation of care it is crucial to understand the perspectives and needs of citizens (WHO) [34]. Citizen engagement will lead to more effective public policies and will lead to an improved quality of care [34–36]. This study therefore made an important contribution to the organization of relocating care.

A strength of this research is the mixed method approach. By means of the questionnaire, we achieved a large reach and it provided an overall picture of citizens' perspective on the subject. Moreover, by means of the citizen platforms, we were able to ask more detailed questions and to expand on the topic. The citizen platforms also offered the opportunity to provide an additional explanation of relocating care. This enabled participants to form an opinion on a potentially difficult or unknown subject for them. In addition, as far as we are aware, this is the first study to conduct extensive qualitative research, not just into the patients' perspective, but from the broader citizen' perspective- on various forms of relocating care. Previous studies have tended to focus solely on the patient perspective [17, 19, 20] or on a specific form of relocating care [13–20].

There are also some limitations to this study. It should be noted that there may be a selective group of citizens in the citizen platforms. Citizens who are in the Dutch Healthcare Consumer Panel might have a special interest in healthcare in general, which could affect their perspectives compared to those who are less interested. They might, for example, be more inclined to consider solutions like relocating care, recognizing its importance. However, the recruitment of participants through the panel allowed us to take a non-probabilistic sample, based on the characteristics of panel members that were already known. During the second citizen platform, this also allowed us to invite people who not wanted to be involved in *the Right Care in the Right Place*, which helped minimize the risk of a biased sample based solely on interest in the topic.

Another aspect to take into account when interpreting the results is that the questionnaire contained open-ended questions. Because of this, respondents are not guided while answering the questions but can formulate their own answers. A disadvantage is that respondents will skip questions more often.

This study did not include relocating care from the hospital to self-care or to care at home. This was a deliberate choice, as such changes involve more significant shifts in care relocation. In this study, we aimed to focus on the smaller steps involved in relocating care. Also, this study focussed on an organisational perspective as a solution to ensure the healthcare system is sustainable in the future. However, there are also several other perspectives. The financial perspective would involve raising taxes or increasing the healthcare insurance premium [4]. There is also the financial organisational perspective, reassessing the boundaries between public and private expenditure. And, finally, the personnel organisational perspective requiring an investment in manpower [5]. While these may provide solutions, they are not discussed in this study.

Some of the conditions mentioned by citizens such as their preference for expertise instead of low costs, may be strongly influenced by the way healthcare is organised and paid for in The Netherlands. Therefore, only general conclusions can be drawn for those countries with a similar organisation of care or countries with a strong primary care, such as the United Kingdom, Belgium, Spain, Portugal, Finland, Estonia, Lithuania, Denmark, or Slovenia [37].

This research has shown that citizens have a positive attitude towards relocating care. However, this research was hypothetical and further research should, therefore, focus on the decisions citizens make in practice. Follow-up research could also shed light on perspectives that were not included in this study, such as the perspectives of HCPs, as they are also important actors in relocating care.

## Conclusion

There seems to be support for relocating care among citizens when certain conditions are met. According to citizens, the challenges faced include training providers so that they have sufficient expertise and providing citizens with knowledge. This would enable them to know which providers to contact and what they can do regarding to self-care. Aspects that emerged as most important within care delivery were the expertise of the provider and, to a lesser extent, the familiarity with the provider. These are also important aspects to maintain or realise when relocating care. Relocating care might not work for patients who are already receiving long-term treatment in a particular place. However, changing the system for new patients could contribute to achieving sustainable healthcare and improving satisfaction among patients. The findings in this study can help to find the best way of organising the relocation of care and, thereby, improve the quality of care.

## Supporting information

**S1 File.**
(DOCX)

## Acknowledgments

We would like to thank the respondents who filled out the questionnaire and the participants who engaged in the citizen platforms. Additionally, we extend our gratitude to Rob Timans, Maaike Horsselenberg, Frank van der Hulst and Anne Brabers, researchers at Nivel, who helped organising and moderating the citizen platforms.

## Author Contributions

**Conceptualization:** L. J. Damen, J. D. De Jong, L. H. D. Van Tuyl, J. C. Korevaar.

**Formal analysis:** L. J. Damen.

**Investigation:** L. J. Damen, J. D. De Jong, L. H. D. Van Tuyl, J. C. Korevaar.

**Methodology:** L. J. Damen, J. D. De Jong, L. H. D. Van Tuyl, J. C. Korevaar.

**Project administration:** J. D. De Jong, L. H. D. Van Tuyl, J. C. Korevaar.

**Supervision:** J. D. De Jong, L. H. D. Van Tuyl, J. C. Korevaar.

**Writing – original draft:** L. J. Damen.

**Writing – review & editing:** J. D. De Jong, L. H. D. Van Tuyl, J. C. Korevaar.

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
