## [Decision Letter · Decision Letter 0]

20 May 2024

PONE-D-22-33488Citizens’ perspectives on relocating healthcarePLOS ONE

Dear Dr. Damen,

Thank you for submitting your manuscript to PLOS ONE. After careful consideration, we feel that it has merit but does not fully meet PLOS ONE’s publication criteria as it currently stands. Therefore, we invite you to submit a revised version of the manuscript that addresses the points raised during the review process.

We look forward to receiving your revised manuscript.

Kind regards,

Andreas Vilhelmsson, Ph.D

Academic Editor

PLOS ONE

Journal Requirements:

2. In the online submission form you indicate that your data is not available for proprietary reasons and have provided a contact point for accessing this data. Please note that your current contact point is a co-author on this manuscript. According to our Data Policy, the contact point must not be an author on the manuscript and must be an institutional contact, ideally not an individual. Please revise your data statement to a non-author institutional point of contact, such as a data access or ethics committee, and send this to us via return email. Please also include contact information for the third party organization, and please include the full citation of where the data can be found.

3. In the ethics statement in the Methods, you have specified that verbal consent was obtained. Please provide additional details regarding how this consent was documented and witnessed, and state whether this was approved by the IRB

Reviewers' comments:

Reviewer's Responses to Questions

**Comments to the Author**

1. Is the manuscript technically sound, and do the data support the conclusions?

Reviewer #1: Yes

Reviewer #2: No

2. Has the statistical analysis been performed appropriately and rigorously? 

Reviewer #1: No

Reviewer #2: No

3. Have the authors made all data underlying the findings in their manuscript fully available?

Reviewer #1: Yes

Reviewer #2: Yes

4. Is the manuscript presented in an intelligible fashion and written in standard English?

Reviewer #1: Yes

Reviewer #2: No

5. Review Comments to the Author

Reviewer #1: Citizens’ perspectives on relocating healthcare.

It is a very interesting and relevant subject area, which is covered in the article Citizens’ perspectives on relocating healthcare.

In which year was the two platforms organised?

Lines: 19, 94, 100: … two citizen platforms were organised in March and April 2022.

Line 128: Citizen platforms were organised on 25 March and 1 April, 2021

The results from the survey were themed, interpreted and discussed. The authors do not show an overview/report which themes emerged from the analysis of the survey.

Comment: The data processing is not described in sufficient detail

Requirements for figure and table text. It is now written as headings.

There must be more supplementary table text. Example Table 4. Now you have to look back in the text to see which/and when rankings were carried out. There is also a lack of explanation for Table 4. How and from what was the result calculated? (Average position (range 1-6)) Comment: The data processing is not described in sufficient detail

Line 268, figure number and text is missing.

Line 380, Is this a claim that came from citizen platform?:

Line 380: So people can ask each other for advice regarding healthcare questions

Line 405, Research has shown that relocating care could amount to a cost reduction of €1,5 billion (10). In addition, by relocating care to e-Health or to self-care, it is expected that fewer health personnel are needed. Comment: There should have been a reflection on the last claim. This is not supported by the data.

Suggests that the article be published, provided that what is missing is described.

Primarily:

• The data processing is not described in sufficient detail

• Reflections on claims, supported by data.

Reviewer #2: Introduction

• Questions in the last paragraph should be removed into the methods section.

Methods

• The first paragraph needs more elaboration. Clarifying the rationale behind this sampling strategy would strengthen the methods section and provide a clearer picture of how the research was conducted. In the study limitations, it was stated that the panel may not be fully representative of Dutch society. Given this limitation, it is imperative to explore how the authors ensured the generalizability of their findings, both qualitatively and quantitatively. This could involve discussing any strategies implemented to mitigate the potential biases introduced by the non-representative sample, as well as considering the broader implications of the findings beyond the sampled population.

• While the response rate for the questionnaire is mentioned (53%), it would be beneficial to discuss efforts made to improve this rate or potential biases introduced by non-response.

• The heading “The three questions in the questionnaire” does not seem appropriate in this section.

• In the first paragraph of data analysis: “The age distribution of the respondents was not quite the same as the composition of the sample. In particular, young people (aged 18-39) responded less often than people aged 40 and over” is the study result and should be moved to the results section. Besides, the low response rate of young people is a study limitation.

• To select participants for the second round of the citizen platform, how did the researchers make sure that having some specific characteristics were related to having negative attitudes towards relocation? Was there any evidence before?

• How did the authors ensure the saturation of data? Were 16 individuals for round 1 and 7 individuals for round 2 enough for data saturation?

• Please briefly describe the content of the topic in the text as well.

• Please include information about the researcher’s or research team’s reflexivity, characteristics and background of the researcher or team, and the nature and length of the researcher’s or team’s relationship with the study participants. How the authors critically examined their own role, potential bias and influence during the formulation of the research questions and data collection?

• In the methods section, please describe how themes and sub-themes were analyzed?

• How the authors ensure of the credibility of their findings?

Results

• The ranking method applied in Table 4 is not clear.

• What do you mean by component C of the citizen platform?

• The results and narrations in the qualitative results sometimes are long. Please consider word limitations in the results section.

6. PLOS authors have the option to publish the peer review history of their article (what does this mean?). If published, this will include your full peer review and any attached files.

Reviewer #1: No

Reviewer #2: No

---

## [Author Response · Author response to Decision Letter 0]

3 Jul 2024

Response to Reviewers

Comments from the reviewers – reviewer 1

Comment 1: In which year was the two platforms organised?

Lines: 19, 94, 100: … two citizen platforms were organised in March and April 2022.

Line 128: Citizen platforms were organised on 25 March and 1 April, 2021

Response to comment 1: We apologize for the confusion. The citizen platforms were organised in 2022. We modified this in the manuscript. 

Comment 2: The results from the survey were themed, interpreted and discussed. The authors do not show an overview/report which themes emerged from the analysis of the survey. 

Response to comment 2: In response, we have now included an overview of the themes that emerged from the analysis of the survey in Appendix D.

Comment 3: Requirements for figure and table text. It is now written as headings. There must be more supplementary table text. Example Table 4. Now you have to look back in the text to see which/and when rankings were carried out. 

Comment 4: There is also a lack of explanation for Table 4. How and from what was the result calculated? (Average position (range 1-6))

Response to comment 3 and 4: We agree with the reviewer that more explanation could be provided for the tables. The following text parts have been added:

Table 1: “*In the left column, each working form is indicated by a letter for easy reference throughout the rest of the article. This can be seen in the column labeled "Component."

*A citizen platform features a one-day program where citizens participate in interactive assignments and attend expert presentations to develop informed opinions. The components listed in this table are part of the one-day program, specifically focusing on the elements from which the results of this article are derived.“

Table 4: “*Each participant could add one aspect of their choice at a blank card.

“*One of the assignments during the citizen platform involved ordering cards with various aspects of healthcare delivery by importance. Each participant received five cards with predefined aspects and one blank card to add their own aspect. The average ranking position for each aspect was calculated from the 23 individual rankings, and the results are shown in this table.”

Comment 5: The data processing is not described in sufficient detail.

Response to comment 5: We acknowledge the reviewer's feedback regarding the insufficient detail in describing the data processing. In response, we have enhanced the method section by providing additional information about the design, data collection, and data analysis. Furthermore, we have restructured the method section by incorporating more sub-headings and reordering the information to improve accessibility. The following details have been integrated into the article:

“Design

This study employed two distinct methods to address the research questions. Firstly, a questionnaire with three open-ended questions was distributed to citizens to achieve broad participation. Secondly, two citizen platforms were organized to facilitate in-depth discussions. 

The questionnaire was chosen for its ability to reach a large audience, while the citizen platforms allowed for more detailed engagement with a smaller group. The citizen platforms are designed to involve participants in discussions on complex health issues, which they might otherwise find difficult to engage in. In the Netherlands, this method was developed by Nivel, the Netherlands Institute for Health Services Research (Bos, 2024), and was inspired by the UK National Institute for Health and Care Excellence Citizens Council (National Institute for Health and Care Excellence (NICE), 2002). The method features a one-day program where citizens participate in interactive assignments and attend expert presentations to develop informed opinions. The goal is not to reach a consensus but to capture a range of perspectives.”

“Presentations were held by Nivel and the Ministery of Health, Welfare and Sport, to explain more about the challenges of the healthcare system and about The Right Care In The Right Place. During the working sessions, participants were presented with cases of care that could be relocated, helping them visualize the situation. Participants then brainstormed their thoughts on these cases, considered other forms of care that could be relocated, and discussed the conditions under which such relocations could occur.”

“Discussions in smaller groups (component C) were audio-recorded. Additionally, participants documented the findings of each group or individual on paper or flip charts, while Nivel researchers also took notes. The subsequent citizen platform on April 1 was conducted online, allowing for the audio recording of all the discussions, in addition to the documentation made by participants and notes taken by the researchers. 

A member check was conducted by sending a summary of the results to the participants after the citizen platforms were finished. Participants were invited to respond if they believed something was incorrect or missing from the summary.”

“Furthermore, the written assignments and flip charts created by participants were cross-referenced with the audio recordings and researchers' notes to verify the results and add any remarks about information not captured in the audio or notes.”

Comment 6: Line 268, figure number and text is missing.

Response to comment 6: We have added the number and text for this figure. 

Comment 7: Line 380, Is this a claim that came from citizen platform?: Line 380: So people can ask each other for advice regarding healthcare questions.

Response to comment 7: This is indeed a claim that came from the citizen platform. To clarify, we have revised the sentence as follows:

“Participants mentioned that this would enable people to ask each other for healthcare advice more often.”

Comment 8: Line 405, Research has shown that relocating care could amount to a cost reduction of €1,5 billion (10). In addition, by relocating care to eHealth or to self-care, it is expected that fewer health personnel are needed. Comment: There should have been a reflection on the last claim. This is not supported by the data.

Response to comment 8: Relocating care is seen as a solution to make care more sustainable by various organizations, such as the OECD and the Dutch government (Organisation for Economic Co-operation and Development, 2017; Taskforce De Juiste Zorg Op de Juiste Plek, 2018). However, there are no exact numbers on the potential savings in terms of the reduced need for personnel. Nevertheless, organizations view it as a viable solution, and it is therefore important to investigate how citizens perceive this reform of the healthcare system. In response to the reviewers' comment, we have modified the text to focus on the main message:

“It is assumed that reforming healthcare systems with the aim to make better use of resources will make a key contribution to keeping healthcare sustainable. One way to do this is by relocating care to more cost-effective settings (Organisation for Economic Co-operation and Development, 2017). To organise the relocation of care it is crucial to understand the perspectives and needs of citizens (World Health Organization, 2022). Citizen engagement will lead to more effective public policies and will lead to an improved quality of care (Epstein, 2008; Irvin & Stansbury, 2004; World Health Organization, 2022). This study therefore made an important contribution to the organization of relocating care.”

Comments from the reviewers – reviewer 2

Comment 1: Questions in the last paragraph should be removed into the methods section.

Response to comment 1: Thank you for your feedback. We understand the reviewer's perspective. The research question could be placed in either the introduction or the methods section. However, we chose to include it in the introduction because we believe it fits better there. We consider this a matter of preference, and therefore, we have decided to keep the research question in the introduction.

Comment 2: The first paragraph needs more elaboration. Clarifying the rationale behind this sampling strategy would strengthen the methods section and provide a clearer picture of how the research was conducted. In the study limitations, it was stated that the panel may not be fully representative of Dutch society. Given this limitation, it is imperative to explore how the authors ensured the generalizability of their findings, both qualitatively and quantitatively. This could involve discussing any strategies implemented to mitigate the potential biases introduced by the non-representative sample, as well as considering the broader implications of the findings beyond the sampled population.

Comment 3: While the response rate for the questionnaire is mentioned (53%), it would be beneficial to discuss efforts made to improve this rate or potential biases introduced by non-response.

Response to comments 2 and 3: We agree with the reviewer that the method section needed clarification. Therefore, we have included more information about the sampling strategy and the overall method [see response to reviewer 1]. The three questions in the questionnaire were open-ended. Consequently, the research has a qualitative nature. We noticed that we occasionally described it in a more quantitative manner, and we have removed those sections that might have caused confusion. Since this research has a qualitative nature, the aim is to showcase a diversity of opinions rather than to conduct representative research. Some groups may be underrepresented compared to their presence in the Dutch community; however, their opinions were still included in our research. An n of 792 is very high for qualitative research.

We added the following information to the article: 

“Design

This study employed two distinct methods to address the research questions.”

“The questionnaire was chosen for its ability to reach a large audience, while the citizen platforms allowed for more detailed engagement with a smaller group.”

“Inclusion criteria are that people must be aged 18 years and older, live in the Netherlands, and are able to understand Dutch. There are no exclusion criteria. Background characteristics from all panel members, such as age, gender, ethnicity, and highest level of education completed, are assessed at the start of the panel membership. ”

Comment 4: The heading “The three questions in the questionnaire” does not seem appropriate in this section.

Response to comment 4: We agree with the reviewer and removed the words: “the three questions in the”, only the heading “Questionnaire” remains. 

Comment 5: In the first paragraph of data analysis: “The age distribution of the respondents was not quite the same as the composition of the sample. In particular, young people (aged 18-39) responded less often than people aged 40 and over” is the study result and should be moved to the results section. Besides, the low response rate of young people is a study limitation.

Response to comment 5: We have removed the age distribution information from the method section. Since our research has a qualitative nature, a sample size of 792 is quite large; typically, saturation can be achieved with a sample size around 150 (Tran et al., 2016). Moreover, all groups are represented in our study, with no characteristics showing a zero count in Table 2. This diversity strengthens our study, as discussed in the discussion section, rather than posing a limitation.

Comment 6: To select participants for the second round of the citizen platform, how did the researchers make sure that having some specific characteristics were related to having negative attitudes towards relocation? Was there any evidence before?

Response to comment 6: We concur with the reviewer's observation regarding the clarity surrounding the selection criteria for participants and the rationale behind it. In a prior survey conducted among the same panel members, participants were asked about their willingness to engage in The Right Care In the Right Place initiative (van der Hulst et al., 2021). From this survey, it was noted that 302 panel members expressed a preference not to participate. Consequently, these individuals were invited to participate in the citizen platform. Subsequent analysis of the survey data revealed that certain characteristics were associated with a reluctance to engage in The Right Care in The Right Place initiative, particularly among younger individuals and those with lower levels of education. Therefore, the sample was supplemented with younger and less educated individuals. In response to the reviewer’s comment, we have provided additional details to provide clarity on participant selection methodology. Moreover, we restructured this section: 

“Recruitment for the second citizen platform followed a similar approach. However, this time participants were chosen based on their interest of being involved with The Right Care In The Right Place. In an earlier survey panel members were asked to what extent panel members wanted to be involved in The Right Care In The Right Place (van der Hulst et al., 2021). We expected that those wanted to be involved with The Right Care In The Right Place might have different perspectives on relocating care than those less interested to be involved. Therefore, for the second citizen platform, we selected individuals who did not want to be involved in The Right Care In The Right Place. All panel members who did not wanted to be involved (n=302) were invited. The sample was supplemented with young people (aged 18-39) and low-educated people, those characteristics were associated with not wanting to be involved (van der Hulst et al., 2021). On 28 February an invitation was sent to 1,022 panel members, of whom 17 wanted to participate and ten were invited.”

Comment 7: How did the authors ensure the saturation of data? Were 16 individuals for round 1 and 7 individuals for round 2 enough for data saturation? 

Response to comment 7: Based on previous experiences with citizen platforms, 20-30 participants are deemed optimal for a successful citizen platform. Consequently, we invited 30 participants, which should be more than sufficient to achieve saturation. As citizen platforms are relatively new, there are no established guidelines for sample size. However, if we consider the saturation guidelines for interviews and focus groups, our participant numbers should suffice. For interview studies, 12 interviews are typically sufficient to reach saturation (Guest et al., 2006), while focus group studies recommend at least three groups, each with six to eight individuals (Guest et al., 2016). Our citizen platforms meet these criteria. Additionally, we organised two citizen platforms to ensure comprehensive coverage and diversity of opinions. Based on the comment of the reviewer the following sentence has been added to the article:

“Based on previous experiences with citizen platforms, 20-30 participants are deemed optimal for a successful citizen platform.”

Comment 8: Please briefly describe the content of the topic in the text as well.

Response to comment 8: We agree with the reviewer that there could be some more information about the content of topics during the citizen platform in the text. Therefore we have included the following: 

“The days consisted of different activities with breaks in between, in accordance with the Nivel Platform guidelines (Triemstra et al., 2020). Appendix B shows the time schedules of the citizen platforms. Presentations were held by Nivel and the Ministery of Health, Welfare and Sport, to explain more about the challenges of the healthcare system and about The Right Care In The Right Place. Moreover, several working methods were used (Table 1). During the working sessions, participants were presented with cases of care that could be relocated (from the hospital to the GP or from the GP to other HCPs/self-care), helping them visualize the situation. These cases were compiled and drafted in collaboration with a GP to ensure realism. Participants then brainstormed their thoughts on these cases, considered other forms of care that could be relocated, and discuss

---

## [Decision Letter · Decision Letter 1]

12 Aug 2024

Citizens’ perspectives on relocating healthcare

PONE-D-22-33488R1

Dear Dr. Damen,

We’re pleased to inform you that your manuscript has been judged scientifically suitable for publication and will be formally accepted for publication once it meets all outstanding technical requirements.

Kind regards,

Andreas Vilhelmsson, Ph.D

Academic Editor

PLOS ONE

Additional Editor Comments (optional):

Reviewers' comments:

Reviewer's Responses to Questions

**Comments to the Author**

1. If the authors have adequately addressed your comments raised in a previous round of review and you feel that this manuscript is now acceptable for publication, you may indicate that here to bypass the “Comments to the Author” section, enter your conflict of interest statement in the “Confidential to Editor” section, and submit your "Accept" recommendation.

Reviewer #1: All comments have been addressed

2. Is the manuscript technically sound, and do the data support the conclusions?

Reviewer #1: Yes

3. Has the statistical analysis been performed appropriately and rigorously? 

Reviewer #1: N/A

4. Have the authors made all data underlying the findings in their manuscript fully available?

Reviewer #1: Yes

5. Is the manuscript presented in an intelligible fashion and written in standard English?

Reviewer #1: Yes

6. Review Comments to the Author

Reviewer #1: (No Response)

7. PLOS authors have the option to publish the peer review history of their article (what does this mean?). If published, this will include your full peer review and any attached files.

Reviewer #1: No

---

## [Editor Report · Acceptance letter]

19 Aug 2024

PONE-D-22-33488R1 

PLOS ONE

Dear Dr. Damen, 

I'm pleased to inform you that your manuscript has been deemed suitable for publication in PLOS ONE. Congratulations! Your manuscript is now being handed over to our production team.

Kind regards, 

on behalf of

Dr Andreas Vilhelmsson 

Academic Editor

PLOS ONE